# Potentiating the Benefits of Melatonin through Chemical Functionalization: Possible Impact on Multifactorial Neurodegenerative Disorders

**DOI:** 10.3390/ijms222111584

**Published:** 2021-10-27

**Authors:** Annia Galano, Eduardo G. Guzmán-López, Russel J. Reiter

**Affiliations:** 1Departamento de Química, Universidad Autónoma Metropolitana-Iztapalapa, San Rafael Atlixco 186, Col. Vicentina, Iztapalapa C.P., Mexico City 09340, Mexico; eggl@ciencias.unam.mx; 2Department of Cellular and Structural Biology, UT Health Science Center, San Antonio, TX 78229, USA

**Keywords:** multifactorial diseases, multifunctional drugs, antioxidant, neuroprotection, free radical scavengers, metal chelators, oxidative stress, DNA repair

## Abstract

Although melatonin is an astonishing molecule, it is possible that chemistry will help in the discovery of new compounds derived from it that may exceed our expectations regarding antioxidant protection and perhaps even neuroprotection. This review briefly summarizes the significant amount of data gathered to date regarding the multiple health benefits of melatonin and related compounds. This review also highlights some of the most recent directions in the discovery of multifunctional pharmaceuticals intended to act as one-molecule multiple-target drugs with potential use in multifactorial diseases, including neurodegenerative disorders. Herein, we discuss the beneficial activities of melatonin derivatives reported to date, in addition to computational strategies to rationally design new derivatives by functionalization of the melatonin molecular framework. It is hoped that this review will promote more investigations on the subject from both experimental and theoretical perspectives.

## 1. Introduction

Chemistry is a science for which every detail matters. Thus, every structural modification, regardless how small, will have consequences on the behavior of chemicals. The rational design of new molecules with potential pharmacological benefits is based on this idea. In this context, a highly challenging task is to find single chemical entities with multiple biological activities, because they are expected to have better success when used in the treatment of complex (multifactorial) diseases. Such chemicals are known as multifunctional drugs (MFD), and are also referred to as magic bullets or master keys due to their improved therapeutic effects and fewer side effects. They are also referred to as dual-mechanism, dual-ligand, bifunctional, multimechanistic, multimodal, pan-agonist, multipotential, pluripotential, and multiple-ligands [1].

Melatonin (*N*-acetyl-5-methoxytryptamine, Figure 1) is an amazing molecule with multiple proven benefits. It not only regulates circadian and seasonal rhythms [2,3,4], but also possesses many other functions in living organisms [5], as summarized in Section 6. Therefore, it is logical to think that the molecular framework of melatonin is an excellent candidate for slight modification to obtain new molecules with even wider benefits. Such research is usually conducted with the premise that, because the structural changes are small, the new derivatives will retain most of the activity of the parent molecule. Although this assumption may seem naïve, nature itself has proven that there are molecules that can be considered as relatives of melatonin that exhibit similar benefits and even improved properties for specific purposes. These are reviewed in Section 7.1.

The main goal of this review is to discuss, at least partially, the beneficial activities of melatonin derivatives reported to date; to consider some of the characteristics of neurodegenerative disorders and why these derivatives may be useful in treating them; and to put into perspective the current stage of development of medical drugs based on melatonin. It is hoped that this review will promote more studies in this field from diverse areas of expertise, which are essential in the development and application of chemicals with pharmacological potential.

## 2. Multifactorial Diseases

Most of the chronic and progressive diseases that lead to high morbidity and mortality are multifactorial in nature and involve the accumulation of biological disfunctions that result in diverse tissue and organ damages [6]. Some of these are cardiovascular, cerebrovascular and neurodegenerative disorders, and immune-, metabolic-, and infection-induced diseases [7,8,9,10,11,12,13,14,15]. Many factors, both genetic and environmental, can contribute to the onset and development of multifactorial diseases (MD, Figure 1).

### Neurodegenerative Disorders

This review is focused on the potential benefits of melatonin derivatives in a particular type of MD: neurodegenerative disorders (NDD). There are more than 600 of these [16], including Alzheimer’s (AD), Parkinson’s (PD), and Huntington’s (HD) diseases and amyotrophic lateral sclerosis (ALS). It has been noted that “As the global population ages and the number of individuals expected to develop neurodegenerative conditions increases, the search for an effective cure is becoming progressively more urgent” [17] (p. 1222).

NDD are among the most enigmatic and problematic health disorders, mainly due to their multifactorial nature. Although each has its own molecular mechanisms and clinical manifestations, some general pathways have been recognized [18,19] (Figure 2).

The chronological hierarchy of the events associated with the underlying causes leading to NDD are not fully understood yet [18]. However, the gathered evidence strongly indicates that one can potentiate the other, leading to a self-sustaining cycle that ultimately provokes neuronal death processes [20].

Based on the multifaceted nature of NDD, efficient therapeutic drugs for treating them should be capable of targeting and regulating several pathological aspects, including deactivation of oxidants and metal chelation. Contrary to this expectation, the most common pharmacological research in the past dealt with compounds that have selective activity against specific molecular targets. Although these are effective against diseases with one prevalent alteration, their effectiveness against the majority of multifactorial diseases has been rather disappointing [6]. Thus, there is a new paradigm in drugs designed for the treatment of multifactorial diseases in general and NDD in particular: multifunctional drugs (MFD).

## 3. Multifunctional Drugs

It has been noted that, because NDD involve a large variety of cellular and biochemical changes, the one-molecule one-target strategy is not adequate for treating them [19]. It was also hypothesized that, because many NDD present similar neuronal disorders, it is possible that a single entity may be potentially used for more than one of these illnesses.

The perspective of such a tactic is promising. The evidence gathered to date indicates that MFD not only have better success at modulating complex diseases, but also do not increase side-effects. Some of the reported advantages of MFD over drug combinations are [1,6,17,18]:Both palliative and disease modifying actions;Additive or synergistic therapeutic responses;Reduced risk of drug-drug interactions;Improved drug characteristics (for example ADME properties);More predictable pharmacokinetics and pharmacodynamic relationships;Prolonged duration of effectiveness;Simplified therapeutic regimen;Lower costs.

It has been proposed that “drug design for chronic diseases might be established based on the rational assembly of multiple chemical groups that are putative ligands for the selected cellular macromolecule targets respectively responsible for root cause and symptoms/signs, and therefore generates the desired multi-target effect” [6] (p. 8).

In the particular case of NDD, the multifunctionality of the desired therapeutic one-molecule multiple-target chemicals should include some of the following effects: (i) inhibition of acetylcholinesterase (AChE); (ii) inhibition of monoamine oxidase (MAO); (iii) inhibition of catechol O-methyltransferase (COMT); (iv) antioxidant behavior; (v) free radical scavenging activity; and (vi) metal chelating power [1,21,22,23,24,25,26,27,28,29,30,31,32,33,34].

## 4. Oxidative Stress

Oxidative stress (OS) has been called the chemical silent killer [35] because it does not produce apparent symptoms and there are no routine tests to detect it yet. Thus, the OS-related damage frequently occurs before the affected person becomes aware of it. In addition, OS can be potentiated by physiological and environmental factors, including physical or mental stress, infections, aging, pollution, radiation, cigarette smoke, and many others [36,37,38,39,40,41,42,43,44,45,46,47,48,49]. There is compelling evidence on the role of OS in the onset and development of a large number of diseases. A few examples are cancer [50,51,52,53], cardiovascular diseases [54,55,56,57,58,59,60,61], diabetes [62,63,64], and neurodegenerative disorders including PD and AD diseases, memory loss, multiple sclerosis, and depression [65,66,67,68,69,70,71,72,73].

More than twenty years ago, the role of reactive oxygen species (ROS) in NDD was proposed to be as important as that of microorganisms in infectious diseases [74]. However, there has been a debate about whether oxidative stress (OS) is a cause or a consequence of the neurodegenerative cascade [75]. Today, there is almost a consensus that OS is an early event of neurodegeneration and one of the major factors of NDD [18].

Neuronal tissue is particularly sensitive to OS. The imbalance in pro-oxidant vs. antioxidant homeostasis in the nervous central system results in the production of ROS, which are involved in the initiation or propagation of radical chain reactions [18]. In AD, PD, HD, and ALS, most biomolecules, including lipids, proteins, and DNA, are oxidatively damaged [76]. Nonetheless, the administration of antioxidants as a treatment has been characterized as being too simplistic, and several clinical studies have demonstrated that they have only modest success in the treatment of NDD [77].

It is possible that antioxidants with other protective effects against this kind of disease may have a better impact. Although transition metals are crucial to biological processes, alterations in their homeostasis increase free radical production, which is frequently catalyzed by traces of redox active metals such as iron, zinc, and copper [78,79]. In addition, metal ions promote fibril generation and deposition [80] and metal-induced OS is associated with mitochondrial dysfunction [81,82,83,84]. Thus, a compound capable of exerting both radical scavenging activity and metal-chelating power may be a better protector against OS than a molecule with only one of these properties [18]. Such a compound may also be a good antioxidant candidate to fight NDD.

## 5. Chemical Antioxidants

The word “antioxidant” involves a large variety of possible actions. Thus, antioxidants can be classified according to their means of action against OS:Primary antioxidants (Type I, chain breaking or free radical scavengers): directly react with free radicals yielding less reactive species that are unable to damage biological targets, or end the radical chain reaction.Secondary antioxidants (Type II, or preventive): exert their protection by chemical routes that do not involve direct reactions with free radicals, such as metal chelation, absorption of UV radiation, deactivation of singlet oxygen, repair of primary antioxidants, and decomposition of hydroperoxide into nonradical species.Tertiary antioxidants (Type III, or fixers): capable of repairing, mainly through H or electron transfer, biomolecules that are oxidatively damaged and restore their pristine structures.Multifunctional antioxidants (Type IV, or versatile): can combine more than one of the above-mentioned means of action, or one of them with enhancing enzymatic protection or restoring pathways in the endogenous antioxidative defense system.

Based on this classification, the latter have higher potential to have beneficial effects on the treatment of NDD.

## 6. Melatonin

Melatonin is a ubiquitous and versatile molecule. In addition to being produced by the pineal gland, it is also found in many other organs, including the skin, retina, cerebellum, kidneys, liver, pancreas, and ovaries [85,86,87,88,89,90]. It has immune-enhancing and anti-inflammatory properties [91,92,93], can inhibit cancer progression [94,95,96], and plays homeostatic roles in the mitochondria [97,98,99].

Melatonin is particularly efficient as an antioxidant [100,101,102]. There are several reasons for its exceptional performance in such a role:Melatonin has very low toxicity [103].Melatonin can cross physiologic barriers easily. This is because of its size and solubility (partially soluble in water and highly soluble in non-polar aprotic solvents, including lipids [104,105]).Metabolisms do not cause a decline in melatonin’s protection against OS, because its metabolites are capable of offering the same kind of protection [106,107,108].

The combined protection of melatonin and its metabolites is widely known [109] and usually referred to as a cascade-like protection [110,111,112]. This collective antioxidant capacity (AOC) implies that melatonin can deactivate more than one equivalent of oxidants. This is a multifaceted process that involves free radical scavenging processes (Type I AOC) and metal chelation (Type II AOC) [113]. It has been proposed that the collective action of melatonin and its metabolites takes place in a “task-division” manner (Figure 3). Some members of this family are expected to perform particularly well as free radical scavengers, whereas others (including melatonin itself) would be more efficient as metal chelators [114].

Studies have also shown that melatonin and its metabolites can protect from OS as Type III antioxidants, particularly by preventing DNA damage. The shielding effect of these compounds against the structural modification of DNA, and the consequent health issues, is believed to be mediated by their capability to repair OS-induced damage to DNA sites. The repairing process was hypothesized to involve various chemical routes. The guanine-centered radical cations are expected to be repaired via electron transfer, at very high rates; C-centered free radicals (in the sugar moiety of 2′-deoxyguanosine), via formal H atom transfer; and OH adducts (in the imidazole ring), by a two-step route: an H atom transfer followed by dehydration [115].

There is abundant evidence supporting the role of melatonin as a protector against OS and derived damages [101,102,116,117,118,119,120,121,122,123,124]. In fact, such a role has been proposed as responsible for many of its numerous benefits for human health [125,126,127,128,129,130,131,132]. It has been even hypothesized that melatonin’s main function in living systems is to prevent them from oxidative damage [133]. The AOC of melatonin acquires particular relevance regarding NDD. The brain is highly susceptible to oxidation and the BBB prevents many chemicals from entering the region. Melatonin is an efficient antioxidant generated in situ and released into the cerebrospinal fluid [134]. Therefore, melatonin is available when needed and capable of protecting against OS. In fact, there is evidence that melatonin has beneficial effects on OS-related disorders that affect the brain [135]. Some examples are Parkinson’s and Alzheimer’s diseases [136,137,138,139,140,141,142,143]. This does not mean that melatonin’s protection against OS is exclusive to the brain. It also protects other organs from this chemical stress, including the stomach, the heart, the liver, and the skin [144,145,146,147,148,149,150,151,152,153,154].

Melatonin is capable of reducing the molecular damage arising from high amounts of free radicals in vivo [155]. It has been reported that melatonin is also capable of scavenging diverse oxidants. Some of these are: ^•^OOCCl_3_ [156], ^•^OH [157,158,159,160], ^•^OR [161,162], ^•^NO [163,164], and ^1^O_2_ [165,166]. Melatonin protects low-density lipoprotein from oxidation [167] and binds copper, which prevents lipid peroxidation induced by this metal [168]. Based on the chelating ability of melatonin, it has been suggested that the neuroprotection (and also the AOC) of this molecule is a consequence of its ability to remove redox metals from the CNN [168]. This hypothesis is supported by other findings. Melatonin can act as a ligand, forming complexes with diverse metals, including not only copper, but also iron, cadmium, aluminum, zinc, and lead [169]. Melatonin significantly lessens the production of free radicals induced by the interaction of Cu(II), Fe(II) Al(III), Mn(II), and Zn(II) with the β-amyloid peptide [170]. In addition, melatonin is also capable of counteracting the oxidative damage to proteins induced by Cu(II)/H_2_O_2_ mixtures [171]. The protective effects of melatonin against molecular damage catalyzed by redox metals were recently thoroughly reviewed [172]. Thus, melatonin itself can be classified a multifunctional antioxidant, and a multifunctional molecule in general (Figure 4).

## 7. Derivatives

### 7.1. From Nature

Among the naturally-occurring melatonin derivatives, its metabolites and precursors stand out (Figure 2). These are all involved in the tryptophan enzymatic metabolism. The direct precursor of melatonin is serotonin (also referred to as normelatonin). It is produced from 5-hydroxytryptophan in mammals and from tryptamine in plants, which are the immediate products of tryptophan in the respective routes.

*N*^1^-acetyl-5-methoxykynuramine (AMK), *N*^1^-acetyl-*N*^2^-formyl-5-methoxykynuramine (AFMK), cyclic 3-hydroxymelatonin (c3OHM), 5-methoxytryptamine (5MTA), and 6-hydroxymelatonin (6OHM) are melatonin’s metabolites. These are produced not only by metabolic routes, but also from the melatonin-mediated free radicals’ detoxification. It is assumed that c3OHM is formed as a result of the reaction between melatonin and the hydroxyl radical [173]. AFMK is yielded when c3OHM further reacts with other free radicals. The virtuous AOC process continues and AMK is a product derived from the radical scavenging reactions of AFMK [174]. By comparison, in mammals, 2-hydroxymelatonin (2OHM) and 4-hydroxymelatonin (4OHM) are the result of the melatonin’s metabolism induced by UV radiation [175]. In plants, 2OHM is enzymatically produced and the most abundant metabolite of melatonin [176].

The AOC of melatonin-derived compounds that are naturally produced has been widely demonstrated. The different manifestations of this activity are summarized in Table 1. They clearly show that all of these compounds are versatile antioxidants.

### 7.2. From the Lab

The appealing properties of melatonin have inspired the search for synthetic derivatives of this molecule intended for a variety of purposes, including AOC protection (Figure 3) [217,218]. The main strategy behind the search has been to modify the indole ring by functionalization at different sites. Some of the groups that have been used in the structural modifications of the melatonin framework are benzoyl and phenyl groups [219,220]. The produced derivatives were demonstrated to surpass melatonin in their anticonvulsant, anxiolytic, and sedative properties. The anti-inflammatory effects of acetyl- and benzoyl-melatonin were reported to be higher than those of melatonin [221]. In addition, they all were found to exhibit significant AOC [222].

The AOC of several melatonin derivatives obtained from functionalization with the sulfhydryl group has been tested, and some of these were proven to have improved antioxidant protection compared to melatonin [223]. Melatonin derivatives containing retinamide [224], 2-phenylindole [225], indole-3-propionamides [226], and N-methylindole plus hydrazide/hydrazone [227] moieties were reported to inhibit lipid peroxidation. Chloroindole hydrazide/hydrazone melatonin derivatives were found to be highly efficient as free radical scavengers [228]. Modifications in the methoxy and acylaminoethyl groups of melatonin have been also pursued as promising synthetic routes to produce compounds with enhanced AOC compared to melatonin [229]. Indole amino acid derivatives of melatonin have been produced and tested for AOC. Their performance as DPPH (2,2-difenil-1-picrylhydrazyl) scavengers was found to be similar to that of melatonin, whereas their efficiency as inhibitors of lipid peroxidation was higher [230].

Thus, there is a large amount of evidence supporting the hypothesis that modest chemical modifications (functionalization) to the melatonin’s molecular framework may lead to the discovery of novel compounds with a wide variety of desired properties. Moreover, they may retain and even surpass the health benefits of the parent molecule.

### 7.3. Computational Designed

When designing multifunctional drugs, three key aspects must be addressed:(i)Building the candidates;(ii)Sampling the search space;(iii)Evaluating their potential for the desired purpose.

The sampling and evaluation stages should simultaneously include: (a) positive design controls that help identifying the chemical space where there is a higher probability of finding drug-like molecules, and (b) negative design controls (‘taboo zones’) that define unwanted properties and/or structures [231]. A unified quantity, allowing direct comparisons among a large number of candidates, is also required. For this purpose, a weighted scoring function is frequently used. This consists of a summation of terms (as many as needed) weighted by factors that are proportional to their relative importance: f(p) = w1p1 + w2p2 + … Part of the challenge here is setting such weighting factors. It appears worthwhile to mention that properties considered in the terms of the summation are diverse and go further than the binding affinity. Some important properties to consider are ADME (absorption, distribution, metabolism, excretion), toxicity, and synthetic accessibility.

A computational protocol known as CADMA-Chem has been recently proposed for the design of antioxidants with multifunctional behavior [232]. This is based on a diversity of chemical properties and involves building a modest number of candidates (a few hundred). The approach used to build them is to add functional groups (1 to 3) to a molecular framework that is already known for a desired purpose. The computational strategies used to design MFD can be roughly grouped into library screening or rational design [17]. The CADMA-Chem protocol is similar to the latter. The difference is that classical rational design combines two (or more) molecular scaffolds (with the target properties) into a single molecular entity, whereas CADMA-Chem slightly modifies a single molecular scaffold (with the desirable pharmaceutical behavior) by adding a few functional groups. This addition is intended to potentiate AOC and improve the physicochemical properties that would allow the candidates to cross lipid barriers (via passive diffusion). In this manner, CADMA-Chem is intended to take advantage of the current knowledge regarding pharmaceuticals that are known to be efficient for the intended purpose, in particular as therapeutic agents in the treatment of OS-related diseases. CADMA-Chem is in line with the three aspects of rational design, as mentioned above. To define negative and positive spaces, this protocol considers AOC, toxicity, ADME properties, and manufacturability.

The hypothesis behind this protocol is to design multifunctional antioxidants with potential as neuroprotectors, although it can be used for other purposes. The desirable properties of the designed molecules are:Free radical scavenging capability;Metal chelation properties (OH inactivating ligand behavior);Low toxicity;Adequate permeation and bioavailability;Non-difficult manufacturability;No pro-oxidant behavior;Efficient for repairing oxidatively damaged biological targets;Inhibition of COMT, AChE and/or MAO.

It is then assumed that a chemical with most of these properties should have neuroprotective effects.

The CADMA-Chem protocol was used to construct melatonin derivatives that are expected to be multifunctional antioxidants [233]. The target multifunctionality consisted of different types of AOC plus possible neuroprotection. By adding different functional groups (-OH, -SH, -NH_2_ and -COOH) to the melatonin scaffold, 116 derivatives were built; 16 with only one functional group, 96 with two functional groups, and four with three functional groups.

The selection score used to sample the search space included eight terms for ADME properties, two for toxicity, and one for synthetic accessibility; each of these had equivalent weight factors. Using this selection score, a first subset (20 candidates) was selected. These were the derivatives with the best drug-like behavior. In a second stage of the search, the pKas of these 20 molecules, in addition to various reactivity indices, were calculated. The reactivity indices were chosen to account for H and electron donor capabilities, which are expected to be good indicators of AOC (type I, free radical scavengers) when involving single electron transfer (SET) and/or formal hydrogen transfer (HAT) chemical routes. Five melatonin derivatives were identified as the most likely candidates to be chemical antioxidants, namely: dM-34, dM-115, dM-38, dM-61, and dM-94 (Figure 4), in that order. The efficiency of each of these, as type I antioxidants, was predicted to be higher than those of Trolox and melatonin.

A different approach was followed in a previous work [234]. To build the candidates, it was considered that the structural features enhancing AOC type I and type II are not necessarily the same. Accordingly, in a first stage of the investigation, two sets of melatonin derivatives were built and tested, one for each type of AOC. The main structural modification in the first set (eight candidates) was to add phenolic OH groups, based on the good free radical scavenging activity of phenolic compounds (including natural melatonin derivatives such as NAS and 6OHM) [113]. The second set of melatonin derivatives (seven candidates) was designed to mimic a particular molecular topology that is assumed to promote metal chelation and the associated type II AOC [234]. In addition, it was proposed that the presence of such topology may be considered as a key structural feature of therapeutic agents in the treatment of AD [235].

The separated analyses of the above-mentioned two sets of molecules allowed identification of the most promising structural features in the designed melatonin derivatives that were assumed to independently promote type I and type II AOC. These features were then combined in a third set (four candidates). These were intended to simultaneously act as free radical scavengers (type I AOC) and redox metal chelators capable of inhibiting OH radical production, via Fenton-like reactions (type II AOC). Two of these were proposed as the most promising candidates as multifunctional antioxidants (Figure 5). The selection was made based on the finding that they are highly efficient as OH inhibitors, via metal chelation, and also excellent for scavenging free radicals.

Further investigations of other properties of the newly designed melatonin derivatives discussed in this section are still needed from both experimental and theoretical approaches. Some of these are to explore their possible pro-oxidant behavior in the absence and the presence of redox metal ions, their efficiency for repairing oxidatively damaged biological targets, and their inhibition potential of COMT, AChE, and/or MAO.

## 8. Summary

Most chronic and progressive health disorders leading to high morbidity and mortality are multifactorial diseases. In the particular case of neurodegenerative disorders, efficient therapeutic drugs for treatment should be capable of targeting and regulating several pathological aspects including AChE-inhibition, MAO-inhibition, COMT-inhibition, antioxidant behavior, free radical scavenging activity, and metal chelating power. Thus, antioxidant molecules are good initial candidates.

Melatonin is an astonishing molecule, but it is possible that chemistry will help to identify new compounds derived from it that may exceed our expectations regarding antioxidant protection and perhaps even neuroprotection. This review briefly summarizes the significant amount of data gathered to date regarding the multiple health benefits of melatonin and related compounds. It also highlights some of the recent directions in the discovery of multifunctional pharmaceuticals intended to act as one-molecule multiple-target drugs with potential use in multifactorial diseases, including neurodegenerative disorders. It is hoped that this review will promote more investigations on the subject from both experimental and theoretical perspectives.

## Data Availability

Not applicable.

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
