# Peer review of "Potentiating the Benefits of Melatonin through Chemical Functionalization: Possible Impact on Multifactorial Neurodegenerative Disorders"

_ijms, 2021, doi:10.3390/ijms222111584_

Round 1

Reviewer 1 Report

The authors aimed to address the issue whether modifing melatonin's molecule could improve its effect in disease, in particular, neurodegenerative disease. They introduce the concept of multifactorial disease in the first part of the paper and subsequently, they discuss the evidence regarding modified melatonin and its potential utility. 

I would suggest that the abstract should reflect more the aim of the study. Also I would suggest minor modifications of the Legends for Figures and include abreviation explanation.

Please include affiliation for the second author

Author Response

The authors aimed to address the issue whether modifing melatonin's molecule could improve its effect in disease, in particular, neurodegenerative disease. They introduce the concept of multifactorial disease in the first part of the paper and subsequently, they discuss the evidence regarding modified melatonin and its potential utility.

I would suggest that the abstract should reflect more the aim of the study. Also I would suggest minor modifications of the Legends for Figures and include abreviation explanation.

RESPONSE: The abstract has been modified as suggested. Legends of figures have been modified to include abbreviation explanation.

Please include affiliation for the second author.

RESPONSE: The affiliation of the second author has been included.

Reviewer 2 Report

The paper presented for review briefly summarizes the available data mainly focused on the antioxidant and neuroprotective activity of melatonin as well as similar properties of selected enzymes in melatonin biosynthesis pathway and melatonin metabolites. These features give melatonin and its derivatives therapeutic potential. According to the Authors, this therapeutical potential can be increased through the use of CADMA-Chem to chemically modify these molecules. The paper may be interesting for readers, as it raises an innovative issue concerning computationally designed molecules in relation to melatonin and related compounds. The reviewed paper may encourage scientists to use computational design technologies to discover more potential therapuetic uses of melatonin. Besides a few minor comments mentioned below (mainly editorial comments) I have no further reservations to the paper.

Computer-assisted design of molecules based on the structure of known ligands, as in the case of multifunctional melatonin, is undoubtedly the future in the search for new drugs, also contributing to reducing costs and reducing the time from the design and synthesis of new compounds to their therapeutic application. However as it is well known, it is not possible for the entire process to take place in silico without the participation of an experimental laboratory. Therefore, I would like to ask the Authors if they have already conducted biological studies using melatonin derivatives as designed by CADMA-Chem? If so, on what in vitro or in vivo models; if not, do the Authors plan such research? Are the results of the pre-clinical studies known with other melatonin derivatives? If so, please provide information on this subject in section 7.3.

Specific comments:

  • According to IJMS requirements please follow the instructions throughout the publication: “In the text, reference numbers should be placed in square brackets [ ], and placed before the punctuation”
  • According to IJMS requirements please follow the instructions for lines 68, 106-109: „For embedded citations in the text with pagination, use both parentheses and brackets to indicate the reference number and page numbers; for example [5] (p. 10). or [6] (pp. 101–105)”
  • Lines 96-97 – shouldn’t it be MFD instead of MD?
  • Lines 111-112 – please expand the abbreviations AChE, MAO and COMT here and then later remove the full name in lines 345-346 using only the abbreviations
  • Line 174 – please expand the abbreviation AOC
  • Line 188 –please use an abbreviation for 6-hydroxymelatonin and 4-hydroxymelatonin and use them in lines 233 and 238
  • Line 256 – please expand the abbreviation AAPH
  • Line 298 – please expand the abbreviation DPPH
  • Please standardize the use of capital or lower case characters for the word "melatonin". In lines 194, 196, 199, 208, 215, 226, 239, 240 capital letters are used, whereas in the rest of the paper lower case characters are used.
  • Scheme 2 – the abbreviation MLT is redundant
  • Line 357 - (absorption, distribution, metabolism, excretion) please transfer to line 317 for ADME
  • Lines 377 – removed N-acetylserotonin and 6-hydroxymelatonin
  • Lines 398, 399 – remove catechol O-methyltransferase, acetylcholinesterase, monoamine oxidase

Author Response

I would like to ask the Authors if they have already conducted biological studies using melatonin derivatives as designed by CADMA-Chem? If so, on what in vitro or in vivo models; if not, do the Authors plan such research? Are the results of the pre-clinical studies known with other melatonin derivatives? If so, please provide information on this subject in section 7.3.

RESPONSE: CADMA-Chem is a recently developed computational protocol. It is inserted in a multidisciplinary project that is still in its first stages. So far, a few candidates have been identified as the most promising ones, using this protocol, and they are in the process of being synthetized. After that, several studies are planed both in vitro and in vivo.

Specific comments:

    According to IJMS requirements please follow the instructions throughout the publication: “In the text, reference numbers should be placed in square brackets [ ], and placed before the punctuation”

RESPONSE: The punctuation has been corrected accordingly.

    According to IJMS requirements please follow the instructions for lines 68, 106-109: „For embedded citations in the text with pagination, use both parentheses and brackets to indicate the reference number and page numbers; for example [5] (p. 10). or [6] (pp. 101–105)”

RESPONSE: The embedded citations have been cited as requested.

    Lines 96-97 – shouldn’t it be MFD instead of MD?

RESPONSE: Yes, its now says MFD.

    Lines 111-112 – please expand the abbreviations AChE, MAO and COMT here and then later remove the full name in lines 345-346 using only the abbreviations.

RESPONSE: The manuscript has been modified accordingly.

    Line 174 – please expand the abbreviation AOC

RESPONSE: The abbreviation has been expanded.

    Line 188 –please use an abbreviation for 6-hydroxymelatonin and 4-hydroxymelatonin and use them in lines 233 and 238.

RESPONSE: the abbreviations have been included.

    Line 256 – please expand the abbreviation AAPH

RESPONSE: The ABTS abbreviation has been expanded.

    Line 298 – please expand the abbreviation DPPH

RESPONSE: The abbreviation has been expanded.

    Please standardize the use of capital or lower case characters for the word "melatonin". In lines 194, 196, 199, 208, 215, 226, 239, 240 capital letters are used, whereas in the rest of the paper lower case characters are used.

RESPONSE: The use of lower case characters was standardized for “melatonin”. The “M” is written in capital letter now only when it starts a sentence, as any other word.

    Scheme 2 – the abbreviation MLT is redundant

RESPONSE: the abbreviation was deleted.

    Line 357 - (absorption, distribution, metabolism, excretion) please transfer to line 317 for ADME

RESPONSE: Done.

    Lines 377 – removed N-acetylserotonin and 6-hydroxymelatonin

RESPONSE: Done.

    Lines 398, 399 – remove catechol O-methyltransferase, acetylcholinesterase, monoamine oxidase

RESPONSE: Done.

Reviewer 3 Report

This review manuscript will reach to acceptable quality after my minor concerns.

Concerns

The authors need to add an illustrated figure of melatonin protective functions as one of multifunctional drugs in the chapter 6 melatonin.

I believe that there is no 100% drug in the world without disadvantages. What is the disadvantage of melatonin therapy as a multifunctional drug?

Author Response

The authors need to add an illustrated figure of melatonin protective functions as one of multifunctional drugs in the chapter 6 melatonin.

RESPONSE: The requested figure has been included in the manuscript (current figure 4)

I believe that there is no 100% drug in the world without disadvantages. What is the disadvantage of melatonin therapy as a multifunctional drug?

RESPONSE: We absolutely agree, but as far as we know there are no reports in the literature reporting the disadvantage of melatonin therapy as a multifunctional drug. This is, actually, an aspect to explore in the future.